# Patient-specific computational simulation of coronary artery bypass grafting

Wei Wu[1,2], Anastasios Nikolaos Panagopoulos[1], Charu Hasini Vasa[1,2], Mohammadali Sharzehee[1], Shijia Zhao[1,2], Saurabhi Samant[1], Usama M. Oguz[1,2], Behram Khan[1], Abdallah Naser[1], Khaled M. Harmouch[1], Ghassan S. Kassab[4], Aleem Siddique[3], Yiannis S. Chatzizisis[1,2]*

1 Cardiovascular Biology and Biomechanics Laboratory, Cardiovascular Division, University of Nebraska Medical Center, Omaha, New England, United States of America, 2 Division of Cardiovascular Medicine, Miller School of Medicine, University of Miami, Miami, Florida, United States of America, 3 Division of Cardiothoracic Surgery, University of Nebraska Medical Center, Omaha, New England, United States of America, 4 California Medical Innovation Institute, San Diego, California, United States of America

* ychatzizisis@icloud.com

## Abstract

### Introduction

Coronary artery bypass graft surgery (CABG) is an intervention in patients with extensive obstructive coronary artery disease diagnosed with invasive coronary angiography. Here we present and test a novel application of non-invasive computational assessment of coronary hemodynamics before and after bypass grafting.

### Methods and results

We tested the computational CABG platform in n = 2 post-CABG patients. The computationally calculated fractional flow reserve showed high agreement with the angiography-based fractional flow reserve. Furthermore, we performed multiscale computational fluid dynamics simulations of pre- and post-CABG under simulated resting and hyperemic conditions in n = 2 patient-specific anatomies 3D reconstructed from coronary computed tomography angiography. We computationally created different degrees of stenosis in the left anterior descending artery, and we showed that increasing severity of native artery stenosis resulted in augmented flow through the graft and improvement of resting and hyperemic flow in the distal part of the grafted native artery.

### Conclusions

We presented a comprehensive patient-specific computational platform that can simulate the hemodynamic conditions before and after CABG and faithfully reproduce the hemodynamic effects of bypass grafting on the native coronary artery flow. Further clinical studies are warranted to validate this preliminary data.

**Data Availability Statement:** All relevant data are within the paper and its Supporting information files.

**Funding:** Funding: National Institutes of Health (R01 HL144690), Dr. Vincent Miscia Cardiovascular Research Fund, Great Plains IDeA-CTR and Center for Heart and Vascular Research. The funders had no role in study design, data collection and analysis, decision to publish, or preparation of the manuscript.

**Competing interests:** Yiannis S. Chatzizisis: Speaker honoraria, advisory board fees and research grant from Boston Scientific Inc., Advisory board fees and research grant from Medtronic Inc., U.S. patent (No. 21072P) for the invention entitled "Patient-specific computational planning of coronary artery bypass grafting", Co-founder of ComKardia Inc. All other authors have no relevant conflict of interests to disclose. This does not alter our adherence to PLOS ONE policies on sharing data and materials.

# 1. Introduction

Coronary artery disease (CAD) is a common condition affecting millions of people worldwide. Treatment of CAD with coronary artery bypass graft surgery (CABG) depends on bypassing severe coronary artery stenoses using arterial or vein grafts. CABG is the most commonly performed cardiac surgery throughout the world [1] and provides survival benefits over percutaneous interventions or medical therapy alone in patients with anatomically complex CAD (Class I recommendation) [2]. The long-term success of CABG depends on the patency of bypass grafts, with 10-year patency ranging from 50% for vein grafts up to 95% for internal mammary artery grafts [3]. Graft failure depends on the graft type and the local hemodynamic conditions of the native coronary artery and bypass graft, among other factors [1]. Compared to arterial grafts, vein grafts are more prone to neointimal hyperplasia, which is the main mechanism of their failure [1, 4]. Flow in the grafted native coronary artery also affects graft (primarily arterial) patency [5]. Non-hemodynamically significant native coronary stenoses allow for greater flow through native circulation, diminishing the flow through the bypass graft, ultimately resulting in graft dysfunction and failure [6].

Although the impact of the severity of native artery stenosis upon graft patency is well recognized clinically and forms the basis for CABG guidelines, these recommendations rely upon visual interpretations of coronary stenoses by invasive angiography that suffers from high inter- and intra-observer variability [1]. Quantitative coronary angiography could potentially limit these effects, but it does not correlate well with the hemodynamic significance of a lesion [7]. Conventionally, angiographic stenosis of greater than 70% luminal diameter is considered anatomically significant. However, over 50% of these lesions are not associated with myocardial ischemia [8]. Fractional flow reserve (FFR) has been proposed as the invasive gold standard for identifying ischemic lesions [9]. An FFR-guided strategy of percutaneous revascularization using the FFR ischemic cut-off of 0.80 has been shown to be superior to the invasive angiography-guided approach [10]. Interestingly, there is significant heterogeneity regarding the presence of ischemia in angiographically non-critical stenoses (i.e., 50–90%) [11, 12]. Angiographic FFR is an accurate alternative to invasive wire-based indices but requires an invasive approach [13]. Application of computational fluid dynamics (CFD) derived from coronary computed tomographic angiography (CCTA) has emerged as a non-invasive tool to assess the hemodynamic significance of coronary artery disease (FFR$_{CT}$, HeartFlow, Redwood City, California, USA) [14]. Furthermore, CCTA-based CFD has been used for the assessment of local hemodynamics in the native coronary arteries and bypass grafts [15, 16]. Most of the computational investigations of bypass surgery have been focused on the effect of graft type [17, 18], graft configuration [19], and anastomosis geometry [20] on the local hemodynamics. A study investigated the competitive flow using computational fluid dynamics in animal models [18]. Another study performed a parametric computational analysis of a relationship between the degree of coronary artery stenosis and the risk of graft failure [21]. Other studies used a more sophisticated approach to fluid-structure interaction to explore the association of blood flow and stress in bypass grafts and native arteries [22, 23]. There have also been some recent attempts to use patient-specific CFD to study the flow physiology in the stenotic native coronary artery and virtually implanted grafts [24]. The concept of combining non-invasive anatomical imaging (e.g., CCTA) with CFD analysis to computationally test the hemodynamic effectiveness of bypass grafting and assist in surgical planning is enticing and warrants further investigation.

This study had two Aims: **Aim 1**: To describe and test a new patient-specific computational framework that combines CCTA with CFD to calculate the local hemodynamics in native coronary arteries and bypass grafts, and **Aim 2**: To use the tested computational framework of

Aim 1, to perform computational hemodynamic investigations on the: **(i)** Effect of bypass grafting on the local hemodynamics in the native artery, and **(ii)** Hemodynamic interdependence (competitive flow) between the graft and native coronary artery. To demonstrate that our proposed computational framework is applicable in the clinical setting, we performed a multifactorial computational analysis accounting for resting and hyperemic conditions and varying degrees of native coronary artery stenosis. In this work, we focused on the left anterior descending artery (LAD) grafted with left internal mammary artery (LIMA) grafts.

## 2. Methods

### 2.1 Study design

The two Aims of this work are summarized in Fig 1. In Aim 1, we performed multiscale CFD simulations of patient-specific post-CABG models (including the native coronary artery stenoses) under computationally hyperemic conditions. Then, we compared the computational FFR in the grafted LAD derived from CFD with the angiographic FFR (CAAS vFFR, Pie Medical Imaging, Maastricht, Netherlands). The angiographic FFR was used as a surrogate for the invasive wire-based FFR [25]. In Aim 2(i), we 3D reconstructed post-CABG anatomies from CCTA and computationally created focal lumen stenoses with four degrees of severity (mild, moderate, severe, critical) in the LAD (proximal to the LIMA anastomosis; Fig 2). The stenosis of the LAD arteries of Aim 2 was less than 70%, thus exerting minimal impact. Then, in each post-CABG model with the computationally created LAD stenosis, we computationally removed all bypass grafts to reproduce the pre-CABG coronary artery anatomy (Fig 2). Using the computational framework of Aim 1, we compared the local hemodynamics [resting distal coronary pressure to aortic pressure ratio (Pd/Pa) and FFR] in the LAD before and after the LIMA grafting under computational resting and hyperemic conditions. In Aim 2(ii), we used the same post-CABG models as in Aim 2(i) and studied the impact of four different degrees of

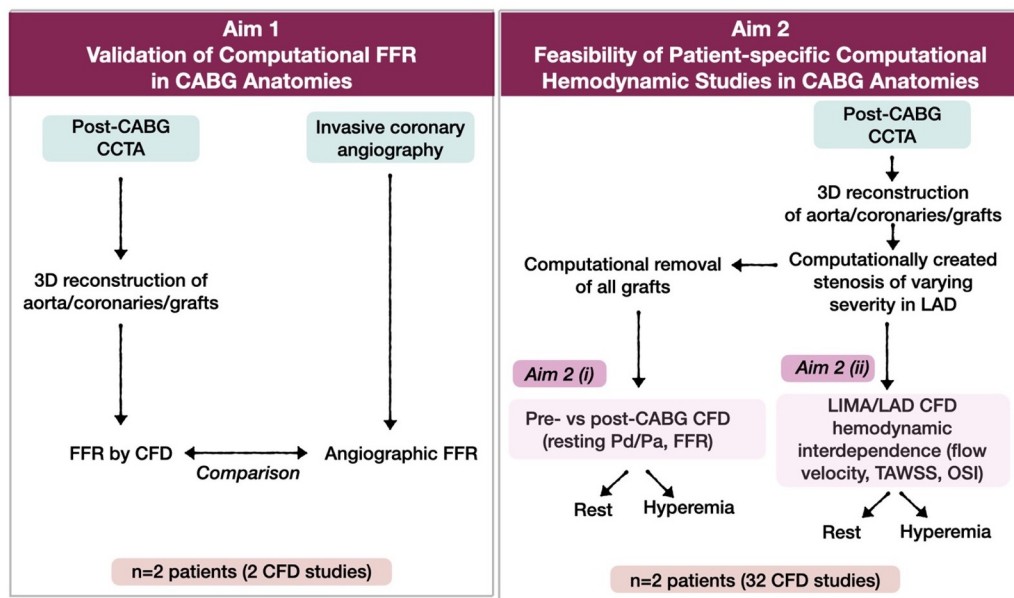

**Fig 1. Study aims.** (CABG: coronary artery bypass grafting, LIMA: left internal mammary artery, FFR: fractional flow reserve, CFD: computational fluid dynamics, CCTA: coronary computed tomography angiography, LAD: left anterior descending artery, TAWSS: time-averaged wall shear stress, OSI: oscillatory shear index).

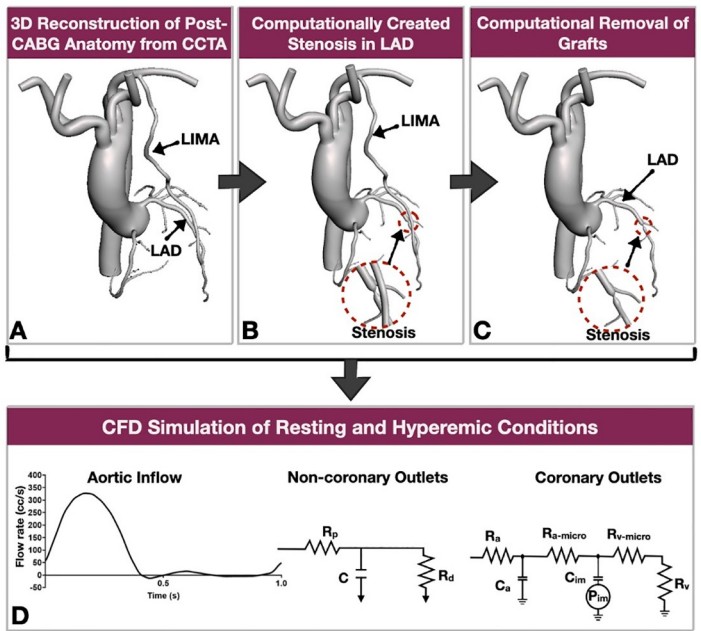

**Fig 2. Illustration of the methodologies employed in Aim 2. A**. The post-CABG anatomy (coronary arteries, aorta and grafts) were 3D reconstructed from CCTA, **B**. In the grafted LAD (proximal to the LIMA anastomosis), we computationally created stenosis of varying severity (magnified in insert), **C**. All the grafts were computationally removed to create the pre-CABG anatomy, **D.** Computational fluid dynamics were performed in pre- and post-CABG anatomies (LIMA: left internal mammary artery, CCTA: coronary computed tomography angiography, LAD: left anterior descending artery, LIMA: left internal mammary artery, $R_p$: proximal vessels resistance, $R_d$: distal vessels resistance, C: capacitance, $R_a$: coronary arterial resistance, $R_{a\text{-micro}}$: coronary venous resistance, $C_a$: Coronary arterial compliance, $C_{im}$: intramyocardial compliance, $P_{im}$: intramyocardial pressure).

LAD stenosis on the flow in the LAD and LIMA under computational resting and hyperemic conditions. In total, in each patient of Aim 2, we computationally simulated 16 different conditions, i.e., 4 degrees of stenosis severity (mild, moderate, severe, critical) x 2 hemodynamic conditions (rest and hyperemia) x 2 CABG conditions (pre-CABG and post-CABG).

## 2.2 Aim 1: Testing of computational FFR in CABG anatomies

### 2.2.1 3D reconstruction and meshing of aorta, coronary arteries and bypass grafts.
Post-operative CCTA imaging data of n = 2 patients were retrospectively acquired from the University of Nebraska Medical Center imaging database (S1 Table). To 3D reconstruct the post-CABG anatomy, the path lines of the ascending aorta, aortic arch, descending thoracic aorta, great vessels, coronary arteries (including their major branches), and bypass grafts were extracted. The lumen boundaries were segmented manually using an open-source software (SimVascular, Stanford, California, USA) [26]. The segmented lumen contours were lofted to create the 3D patient-specific lumen geometries. The 3D reconstructed lumen geometries were discretized into tetrahedral elements using the SimVascular software. Based on a mesh independence analysis that was conducted until a further mesh refinement resulted in less than 5% change in the calculated wall shear stress (WSS), we applied the following mesh resolution: 3 mm in the aorta, 1.5 mm in aortic branches, 0.1–0.3 mm in coronary arteries and grafts. The average total number of elements was 2.1 million.

### 2.2.2 Computational FFR in the grafted LAD.   *a. Computational resting conditions*. We performed multiscale CFD studies in the 3D reconstructed aorta, coronary arteries, and grafts,

as described below. A normal parabolic flow rate was applied at the aortic inlet. The flow rate of a healthy subject (derived from 4D flow MRI [27]) was imposed at the aortic inlet using parabolic velocity mapping (10-mode Fourier). At the noncoronary outlets, a three-element Windkessel model was prescribed to include the resistance (Rp) and capacitance (C) of the proximal vessels and resistance (Rd) only of the distal vessels (Fig 2D). At the coronary outlets, we applied the lumped parameter coronary vascular model, which included the coronary arterial resistance ($R_a$), coronary arterial microcirculation resistance ($R_a$micro), coronary venous microcirculation resistance ($R_{v\text{-}micro}$), coronary venous resistance ($R_v$), coronary arterial compliance ($C_a$), intramyocardial compliance ($C_{im}$), and intramyocardial pressure ($P_{im}$) [28]. Left and right ventricular pressures of a normal adult were applied [29].

The total resting resistance of all outlets was calculated based on the ratio of mean aortic pressure to the mean flow rate. We assumed that only 4% of the cardiac output was distributed to the coronary arteries to calculate the Windkessel boundary conditions at outlets. The modified Murray's law was employed to calculate the resistance of each coronary branch [30]. The total capacitance was distributed proportionally to the outlet areas. The resistance (Rp, Rd. . .) and capacitance (C. . .) of each patient's outlet were iteratively and manually adjusted to match the normal aortic root blood pressure of 120/80 mmHg. These boundary condition values were tuned until the difference to the target pressure of the aorta was less than 5 mmHg. The resistance and capacitance parameters for each patient are listed in S2 Table. The lumen wall was assumed to be rigid (non-deformable) with no slip boundary condition [29]. Blood was assumed as an incompressible Newtonian fluid with a density of 1.06 g/cm$^3$ and a dynamic viscosity of 0.04 Pa-s. Simulations were run for n = 6 cardiac cycles (2,000 time-steps per cycle) to achieve a periodic solution, and the final cardiac cycle was used for the hemodynamic comparison. The detailed solver parameters are listed in the S3 Table. The simulations were run on a computer cluster (452 Intel Xeon E5-2670 2.60GHz 2 CPU/16 cores with 64GB RAM per node) located at the University of Nebraska-Lincoln.

*b. Computational hyperemic conditions.* We computationally modeled the adenosine-induced hyperemia (140 mcg/kg/min) that is applied in invasive FFR measurements [31, 32]. We reduced the coronary and great vessels resistance to 22% and 95% of the resting values, respectively [33]. The FFR was computationally calculated based on the ratio of the pressure at the distal LAD to the pressure at the aortic root under simulated vasodilator-induced hyperemic conditions (S4 Table). In Aim 1, the FFR was calculated distal to the LIMA anastomosis (at the same location with angiographic FFR) and averaged over a zone of 0.5 cm. In Aim 2, the FFR was calculated proximally and distally to the computationally created LAD stenosis.

**2.2.3 Calculation of angiographic FFR in the grafted LAD.** The CFD-derived FFR in the native LAD distal to the LIMA anastomosis was compared to the reference angiography-derived FFR. The angiographic FFR was calculated with the CAAS software (CAAS vFFR, Pie Medical Imaging, Maastricht, Netherlands). Briefly, the LIMA-LAD were semi-automatically segmented in two perpendicular angiographic projections and 3D reconstructed. The systemic aortic pressure was used as input for the automatic calculation of the angiographic FFR. Notably, the post-CABG CCTA and invasive angiogram were acquired within fewer than two months to minimize the effect of disease progression on the calculated hemodynamics with different imaging modalities.

## 2.3 Aim 2: Feasibility of patient-specific computational hemodynamic studies in CABG anatomies

**2.3.1 Computational creation of varying degrees of LAD stenoses.** To assess the impact of bypass grafting on native coronary artery hemodynamics [Aim 2(i)], we

retrospectively collected n = 2 post-CABG CCTAs from the University of Nebraska Medical Center imaging database. The patients' clinical and graft characteristics are summarized in S1 Table. We 3D reconstructed these anatomies, as described in 2.2.1, and then computationally created focal lumen stenosis in the reconstructed LAD with four degrees of diameter reduction, i.e., mild (29%), moderate (50%), severe (68%), and critical (82%) [21]. To eliminate the confounding effect of lesion length on local hemodynamics, all the computationally created stenoses were 5mm in length. The lumen contours proximal and distal to the stenosis were lofted to create the entire stenosis geometry (SimVascular, Stanford, California, USA).

**2.3.2 Computational removal of the grafts from the post-CABG anatomies.** In the post-CABG anatomies, we computationally removed all the grafts from their proximal takeoff (left subclavian artery or aorta) to their distal anastomosis (Fig 2A and 2C). The lumen of the native coronary arteries was lofted using the SimVascular software.

**2.3.3 CFD and calculated hemodynamic parameters.** For the calculation of the local hemodynamic parameters, we used the same computational framework as described in 2.2.2. The average number of elements for pre- and post-CABG was 4.7 and 6.8 million, respectively. The CFD simulations were run in the same computer cluster, as described in section 2.2. We calculated the following local hemodynamic parameters: Pd/Pa, FFR, flow rate, flow velocity, TAWSS, and OSI (S4 Table). The mean pressures used to calculate the resting Pd/Pa and FFR were based on the computationally simulated resting and hyperemic conditions, respectively.

**2.3.4 Study of the hemodynamic interdependence between the graft and native coronary artery.** To assess the impact of different degrees of LAD stenosis on the hemodynamics of LIMA and LAD [Aim 2(ii)], we used the same 3D reconstructed post-CABG models, as described in 2.3.1. For each degree of computational stenosis in the LAD, we performed CFD analysis and calculated the flow velocity in the LAD and LIMA under computationally resting and hyperemic conditions, as described in section 2.2.2.

## 2.4 Ethics statement

University of Nebraska Medical Center IRB. Patient-specific computational planning of coronary artery bypass grafting: A Novel Approach IRB Protocol # 0276-21-FB, Consent form is written.

# 3. Results

## 3.1 Simulated pressure and flow waveforms

The computationally simulated resting and hyperemic conditions in the pre-CABG anatomies resulted in physiologically meaningful pressure and flow waveforms at the LAD distal to the computationally created stenoses. With hyperemia, the mean aortic pressures were reduced by 9.07±0.77 mmHg, in line with literature data [31]. The rest of the modeling parameters, including heart rate, remained unchanged compared to resting conditions. Fig 3 shows a representative example of the pressure and flow waveforms (patient 3). Note that the simulated LAD flow was higher during the diastolic phase of the cardiac cycle compared to systole. The simulated vasodilation-induced hyperemia resulted in a reduction of pressures and an increase in flow rates compared to resting conditions. Higher degrees of LAD stenoses resulted in a stepwise decrease in LAD pressures and flow rates. Overall, these results support the ability of our computational modeling to replicate human coronary artery hemodynamics.

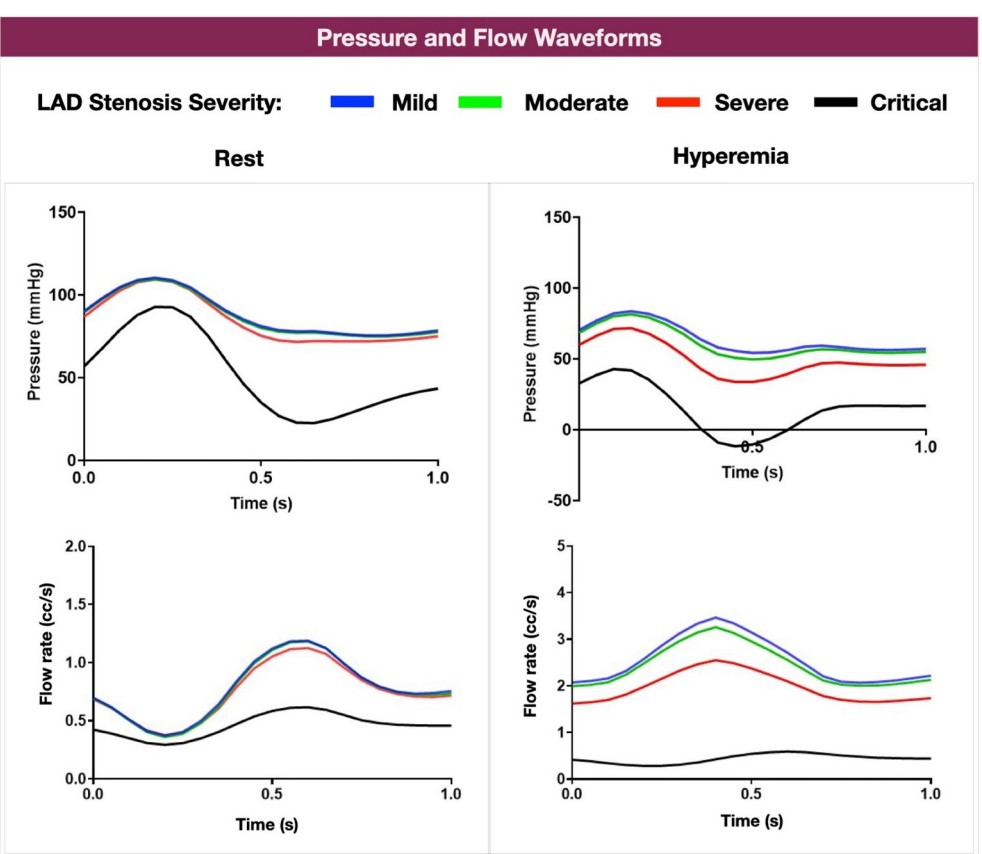

**Fig 3. LAD Pressure and Flow curves across various degrees of LAD stenosis severity pre-CABG at computationally resting and hyperemic conditions (patient 3).** Note the variation of the LAD flow according to the phase of the cardiac cycle and the impact of the increasing severity of LAD stenosis to LAD pressure and flow (LAD: left anterior descending artery).

## 3.2 Computational FFR by CFD vs. angiographic FFR (Aim 1)

As shown in Fig 4, there was a high agreement between the computational FFR derived from CFD and angiographic FFR. Although we used a small number of patients (n = 2), these results were consistent and suggest that our computational CABG platform can faithfully reproduce the local hemodynamic environment of native coronary arteries and grafts.

## 3.3 Discrepancy between anatomical and hemodynamic significance in native (Non-grafted) arteries [Aim 2(i)]

The pre-CABG resting Pd/Pa (resting conditions) and FFR (hyperemic conditions) were computationally calculated proximal and distal to the computationally created LAD stenoses (Fig 5). At both resting and hyperemic conditions, there was an agreement between the anatomical and hemodynamic significance distal to mild (29%), moderate (50%), and critical (82%) lumen diameter stenoses (Fig 5, blue bars). Distal to anatomically severe stenoses (i.e., 68% lumen diameter stenosis), the hemodynamic significance was concordant with the anatomical significance at hyperemic conditions but discordant at resting conditions (Fig 5, blue bars). Notably, on the individual patient analysis, the FFR varied in relation to the anatomical significance in moderate, severe, and critical stenosis (S5 Table). In patient 3, the FFR was below the

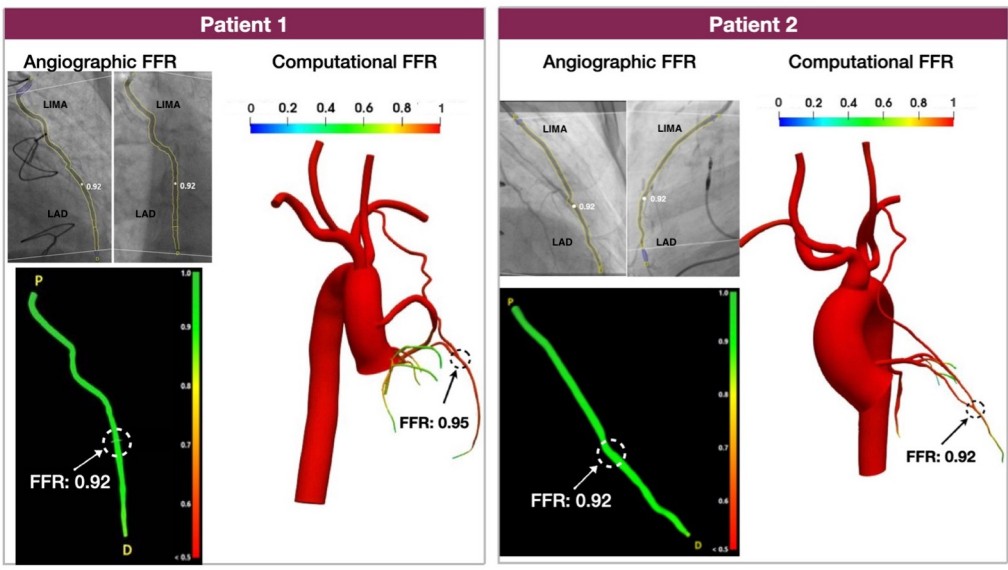

**Fig 4. Comparison of the computationally calculated FFR against angiographic FFR in Aim 1 patients.** Note the high agreement between computational and angiographic FFR. (FFR: fractional flow reserve).

ischemic cut-off of 0.80 in moderate, severe, and critical stenosis, whereas in patient 4, the FFR was below the ischemic cut-off of 0.80 in severe and critical stenosis but above 0.80 in moderate stenosis. This observation from the individual patient analysis is consistent with the established knowledge that anatomical stenoses exhibit varying hemodynamic significance [10] and supports the ability of our computational framework to reproduce these phenomena.

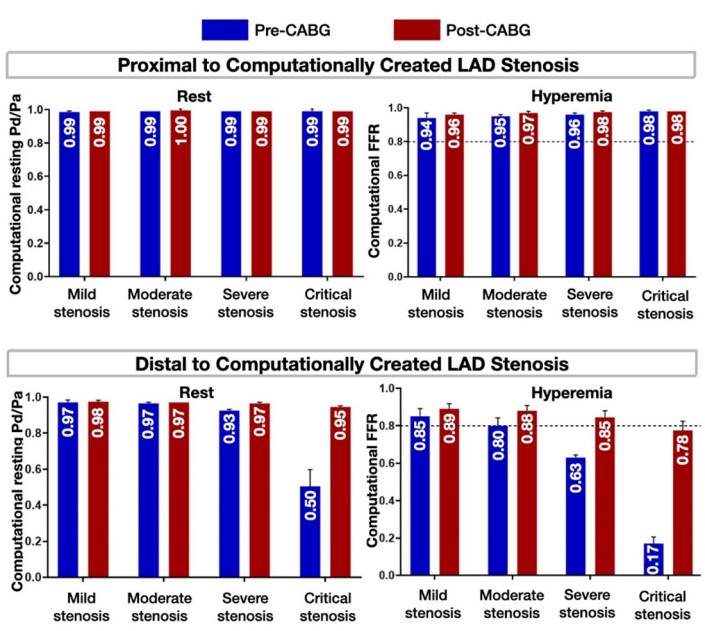

**Fig 5. Effect of LIMA grafting on LAD hemodynamics.** Computational calculation of the pre-and post-CABG resting Pd/Pa (resting conditions) and FFR (hyperemic conditions) proximal and distal to the computationally created stenosis. Values are expressed as mean±SD. Note the normalization of FFR by grafting severe stenosis and near normalization of FFR by grafting critical stenoses (red bars; dashed lines represent the FFR ischemic cut-off of 0.80; Pd/Pa: distal pressure/aorta pressure, FFR: fractional flow reserve, CABG: coronary artery bypass grafting, LAD: left anterior descending artery, SD: standard deviation).

### 3.4 Effects of bypass grafting on native artery hemodynamics [Aim 2(i)]

The post-CABG resting (Pd/Pa) and hyperemic (FFR) indices were computationally calculated proximal and distal to the LAD stenosis (same locations as in 3.3). As expected, proximal to the stenosis, both resting Pd/Pa and FFR were above their ischemic cut-offs for all degrees of lumen stenosis (Fig 5, **red bars**), supporting the correct hemodynamic calibration of our computational platform. Distal to the LAD stenosis, bypass grafting normalized the resting Pd/Pa in critical stenoses (from 0.50 to 0.95). Similarly, bypass grafting normalized the FFR in severe stenoses (from 0.63 to 0.85) and nearly normalized the FFR in critical stenoses (from 0.17 to 0.78) (Fig 5, **blue and red bars**). These results indicate that bypass grafting normalizes the hyperemic pressure drop across anatomically severe or critical native artery stenoses, whereas it has minimal hemodynamic effects on anatomically mild or moderate stenoses.

### 3.5 Hemodynamic interdependence between the graft and native coronary artery [Aim 2(ii)]

**3.5.1 Patient variability of resting flow rates.**   Fig 6A illustrates the flow rates in the LAD and LIMA of patients 3 and 4 at computationally resting conditions for varying degrees of LAD stenosis severity. Note that the point at which the flow of LAD and LIMA equalized (i.e., hemodynamic equilibrium) was different in each patient. In patient 3, for mild and moderate stenosis, the resting flow rate in the LAD was greater than that across the LIMA, whereas the LAD and LIMA flow reached an equilibrium at severe-to-critical stenoses. Beyond that point, the LIMA flow was becoming progressively dominant over the LAD flow as the severity of stenosis increased. In patient 4, the equilibrium between the resting LAD and LIMA flow was achieved at a lesser degree of stenosis compared to patient 3 (moderate-to-severe stenosis vs. severe-to-critical stenosis).

**3.5.2 Variability of flow velocity at rest and hyperemia.**   In each patient, there was a variability of flow in the LAD and LIMA between resting and hyperemic conditions. Fig 6B illustrates the cross-sectional flow velocity contours in the LAD and LIMA of patient 3 at the time of maximum flow rate (i.e., peak diastole) under computationally resting and hyperemic conditions. These results suggest that grafting a non-hemodynamically significant stenosis results in high flow in the native artery competing with the flow through the graft. Conversely, grafting a hemodynamically significant native artery stenosis results in higher flow rates in the graft compared to the native artery.

**3.5.3 Resting TAWSS and OSI.**   Fig 7A illustrates quantitatively the LAD and LIMA lumen area exposed to low TAWSS and high OSI post-CABG under computationally resting conditions (patient 3). In critical LAD stenosis, the LIMA lumen area exposed to low TAWSS was at most one-third of the LIMA area exposed to low TAWSS in non-critical stenosis (i.e., mild, moderate, severe; green bars). The LIMA lumen area exposed to high OSI in critical stenosis was substantially smaller than the LIMA area exposed to high OSI in mild and moderate stenosis. The opposite patterns of TAWSS and OSI were observed in the LAD (red bars). Fig 7B illustrates qualitatively the abovementioned TAWSS and OSI patterns in the LIMA and LAD of patient 3.

## 4. Discussion

The current study described and tested a novel application of CFD for accurate patient-specific assessment of hemodynamics in CABG. Our computational methodology was tested against angiographic FFR performed in clinical cases with varying anatomical complexity. Furthermore, this study demonstrated the feasibility of using patient-specific simulations to study the hemodynamic interdependence of native arteries and grafts after CABG. The novelty in our

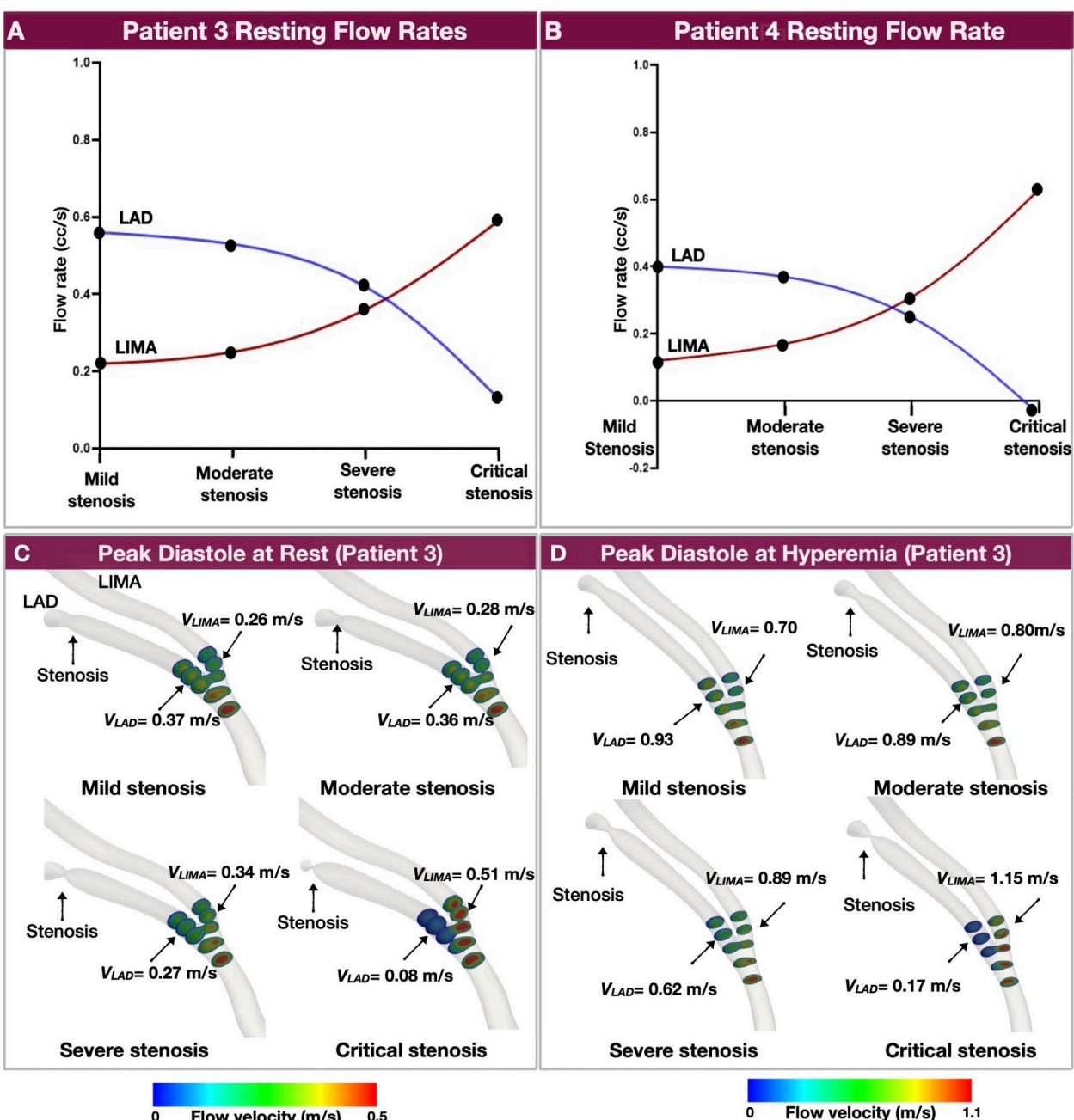

**Fig 6. Hemodynamic interdependence between LAD and LIMA. A and B.** Note the transition of resting flow dominance from the LAD to LIMA as the severity of LAD stenosis increases. **C and D.** Cross-sectional flow velocity contours across the LAD and LIMA for varying degrees of computationally created stenosis at peak diastole under computationally resting (**C**) and hyperemic (**D**) conditions for patient 3. Note that as the severity of the LAD stenosis increases, the dominant flow shifts from the LAD to LIMA. The transition point from the LAD flow dominance to LIMA flow dominance occurs with severe stenosis at rest and moderate stenosis at hyperemia (LAD: left anterior descending artery, LIMA: left internal mammary artery).

work is that we repurposed existing computational modalities and technologies (CCTA, CFD, FFR) to build a new framework and clinical application that could serve as a clinically impactful tool for effective surgical planning based on non-invasive hemodynamic assessment of coronary disease.

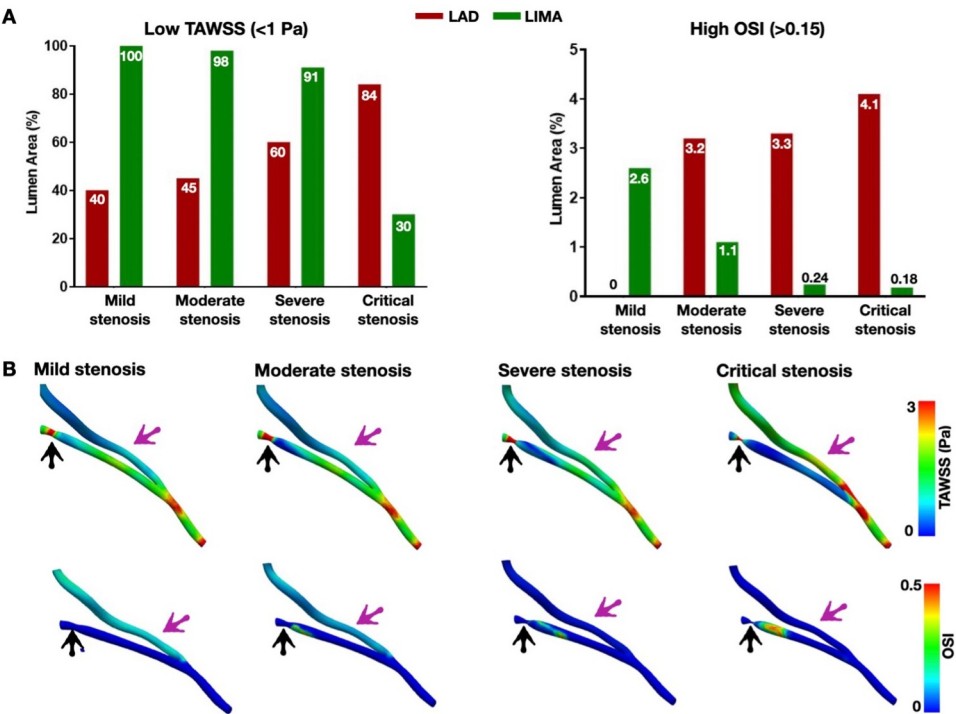

**Fig 7. Representative example of low TAWSS and high OSI areas across the LAD and LIMA after CABG at resting conditions (Patient 3). A.** Quantitative results, **B.** Qualitative results. Note that as the degree of LAD stenosis severity increases, the LIMA lumen area exposed to low TAWSS and high OSI decreases (A, green bars), whereas the LAD lumen area exposed to low TAWSS and high OSI increases (A, red bars). Values are expressed as mean±SD (Purple arrow: LIMA, black arrow: LAD stenosis; LIMA: left internal mammary artery, LAD: left anterior descending artery, TAWSS: time-averaged wall shear stress, OSI: oscillatory shear index).

## 4.1 Patient-specific CFD studies of bypass grafting

Previous studies used computational simulations to investigate the anatomical (e.g., graft geometry [20], graft configuration [19], graft type [17, 18]) and CFD parameters [34] that could be associated with graft failure in CABG patients. However, the lack of post-operative data limited the accuracy of these studies [35, 36]. Another study used patient-specific computational simulations to assess the effect of computationally implanted bypass grafts in the native artery flow [24]. However, the accuracy of these computational simulations is questionable, given the lack of validation in a grafted model. In our work, we build upon the current state-of-the-art and repurpose existing CFD knowledge toward a novel patient-specific application of computational simulations in CABG hemodynamics. We presented a comprehensive computational methodology that can faithfully reproduce the physiologic effects of bypass grafting on the native coronary bed. We showed that as the severity of native artery stenosis increases, the flow in the graft increases, resulting in marked improvement of resting and hyperemic native artery flow distal to the LIMA anastomosis (Fig 8). These computational findings aligned with clinical data from intraoperative flow measurements of native arteries and grafts post-CABG [37, 38]. Also, to identify areas with a theoretical predisposition to atherosclerosis in the native arteries and grafts, we calculated the lumen area percentages exposed to pro-atherosclerotic low TAWSS and high OSI using the cut-off values of 1.0 Pa for low TAWSS and 0.15 for high OSI [39]. We found that with worsening stenosis, the LIMA lumen area exposed to low TAWSS was decreasing, and the native lumen area (proximal to LIMA anastomoses) exposed to low TAWSS was increasing. The same pattern was observed with high OSI.

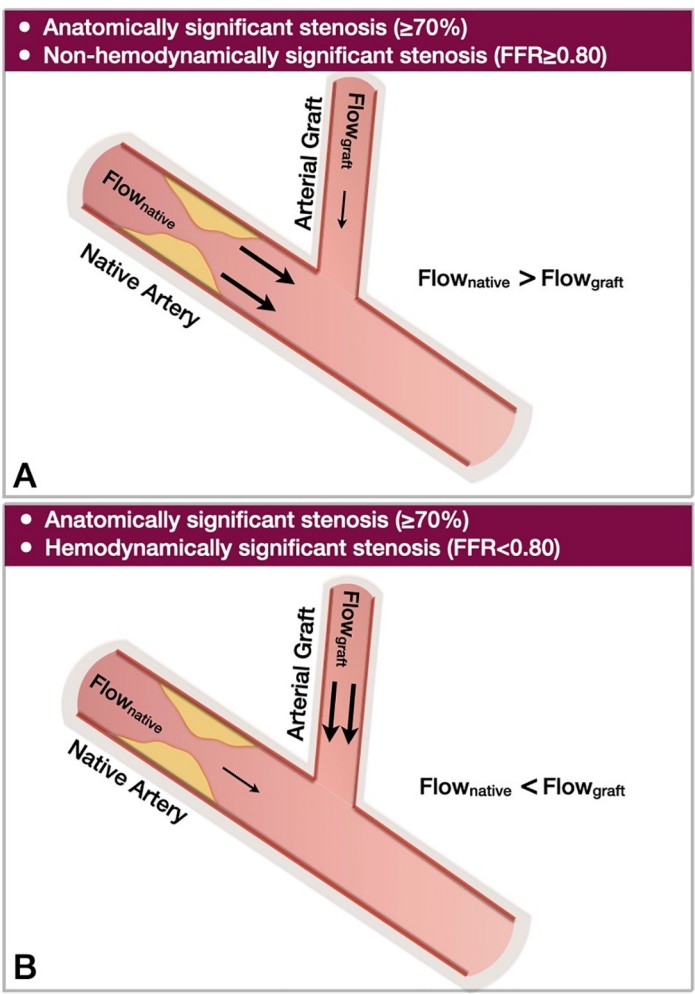

**Fig 8. Schematic presentation of the dominant role of the hemodynamic significance over the anatomical significance of a native artery stenosis on the graft flow. A.** In anatomically significant but non-hemodynamically significant native artery stenoses, the flow through the native artery exceeds the flow through the graft, **B.** In anatomically and hemodynamically significant native artery stenoses, the flow through the graft exceeds the flow through the native artery (FFR: fractional flow reserve).

## 4.2 Hemodynamic interdependence (Competitive Flow) between LAD and LIMA

In this study, we demonstrated that grafting of non-hemodynamically significant LAD stenoses resulted in decreased flow through the LIMA graft and consequently greater exposure of LIMA to low shear stress and disturbed flow compared to LAD. Conversely, grafting of hemodynamically significant LAD stenoses led to more favorable flow patterns in the LIMA graft (i.e., lesser exposure to low shear stress and disturbed flow) compared to the LAD. Even though it is beyond the scope of the current investigation, these findings could provide a hemodynamic explanation of graft failure and native artery disease progression observed with grafting of non-significant lesions [18, 40]. Conditions that lead to low blood flow through the graft could impact the flow-mediated vasodilatation, which is a key factor for the long-term viability of arterial grafts [41, 42]. The native coronary hemodynamics prior to the CABG determine many aspects of the post-CABG hemodynamic outcomes and potentially the clinical outcomes. A

recent meta-analysis of four prospective studies (n = 503 patients) showed that CABG of coronary vessels with hemodynamically non-significant stenoses was associated with an increased risk of graft failure [6]. Notably, the FFR cut-off to predict graft failure was 0.79. Several investigators, including our group, have proposed clinical decision algorithms that incorporate invasive functional studies (e.g., FFR) in the surgical revascularization of patients with multivessel coronary artery disease [1, 43]. To date, clinical studies have shown that the FFR-guided revascularization strategy is superior to the angiography-guided approach in terms of the number of anastomoses and complexity of procedures, however, they have failed to show outcome benefits [43]. Given the variability of invasive functional studies among individuals, the application of a universal cut-off of invasive functional studies to guide revascularization might not be the optimal approach. One could hypothesize that an individualized strategy—like the one presented in this work-based on non-invasive anatomical imaging (e.g., CCTA) coupled with computational simulations could offer accurate predictions about the hemodynamic interdependence between native arteries and grafts, thereby guiding the revascularization plan (type and number of grafts) and potentially improving the long-term graft patency.

### 4.3 Clinical perspectives of computational planning of CABG

Computational planning could be used in virtual (*in-silico*) clinical scenarios, with patient-specific anatomic and physiologic data, to provide surrogate clinical endpoints after computationally adding bypass grafts (i.e., predicted FFR, WSS, OSI, competitive flow in native vessel and graft) that potentially correlate with graft patency and clinical endpoints. Grafting native arteries with less severe stenoses could increase the chances of graft (primarily arterial) failure due to reduced flow through the graft. A clinical study is warranted to establish the connection between simulation surrogate endpoints and clinical endpoints [1]. Although computationally performed CABG could differ in terms of the final graft configuration (e.g., curvature and tortuosity) compared to bypass surgery, a recent study demonstrated that graft geometry in virtual bypass surgery does not have a significant impact on coronary flow [44]. The ability to accurately and non-invasively predict post-CABG hemodynamics may have a significant impact on the clinical practice of CABG by enabling the surgeon to tailor the surgical plan (graft number and type) to each individual patient and achieve the best possible graft patency.

Furthermore, the proposed computational methodology could provide a reliable resource for clinical research. For instance, it could provide the hemodynamic basis for the observation that grafted native coronary arteries exhibit accelerated disease progression compared to non-grafted arteries [45]. Beyond that, our computational platform could serve as an educational vehicle on CABG techniques and could be translated to the surgical revascularization of non-coronary vascular beds (e.g., peripheral arteries). Overall, our study provides a new, hitherto unexplored approach to surgical revascularization, which might have enormous clinical potential.

### 4.4 Limitations

There were several limitations in the current study. First, in Aim 2, the pre-CABG models were based on the post-CABG CCTA by computationally removing all bypass grafts due to the lack of pre-CABG CCTA. Second, we tested our computational FFR against angiographic FFR, which is not the gold standard of invasive functional studies. Nonetheless, the angiographic FFR has been found to correlate very well with invasive FFR and can represent a reliable surrogate of invasive FFR [13]. Third, our computational CABG methodology was tested on a limited number of patients. Our study served as a proof-of-concept for a larger study that could validate our approach further. Fourth, this study focused on the LIMA grafting of the LAD, and extrapolation of these findings to non-LIMA grafts or non-LAD epicardial coronary

arteries remains to be elucidated. Fifth, in Aim 2 of the current study, the pre-existent native epicardial coronary artery disease could confound the effects of the computationally added stenoses on the graft and distal LAD hemodynamics. Since the pre-existent stenoses in the LAD were less than 70%, we believe that they had minimal impact on the computationally created stenoses. Sixth, there were a series of limitations related to our computational methodology: **(i)** The values of the aortic flow rates and ventricular pressures were derived from published data corresponding to healthy individuals and not individual study patients [27, 29]. **(ii)** To reduce the complexity of computational calculations, we used an open-loop model with the same lumped parameters in the pre- and post-CABG simulations [28], **(iii)** In simulations, the vascular wall was considered rigid; however, the capacitance was distributed proportionally to the outlets to capture the global effects of flexible walls [29].

## 5. Conclusions

This work presented a comprehensive patient-specific computational platform that can simulate the hemodynamic conditions before and after CABG and predict the hemodynamic effects of bypass grafting on the native coronary artery flow. Further clinical studies are warranted to validate this preliminary data.

## Supporting information

**S1 Table. Clinical, angiographic, and graft characteristics of the study patients.**
(DOCX)

**S2 Table. Total boundary parameters used in the patients.**
(DOCX)

**S3 Table. Solver parameters used in the CFD simulation.**
(DOCX)

**S4 Table. Hemodynamic parameter definitions.**
(DOCX)

**S5 Table. Computational FFR measurement proximal and distal to the computationally created LAD stenosis pre- and post-CABG.**
(DOCX)

## Author Contributions

**Conceptualization:** Wei Wu, Mohammadali Sharzehee, Aleem Siddique, Yiannis S. Chatzizisis.

**Data curation:** Wei Wu, Anastasios Nikolaos Panagopoulos, Mohammadali Sharzehee, Shijia Zhao, Saurabhi Samant, Usama M. Oguz, Behram Khan, Abdallah Naser, Khaled M. Harmouch, Ghassan S. Kassab, Aleem Siddique, Yiannis S. Chatzizisis.

**Formal analysis:** Wei Wu, Anastasios Nikolaos Panagopoulos, Mohammadali Sharzehee, Shijia Zhao, Saurabhi Samant, Usama M. Oguz, Behram Khan, Abdallah Naser, Khaled M. Harmouch, Ghassan S. Kassab, Aleem Siddique, Yiannis S. Chatzizisis.

**Funding acquisition:** Wei Wu, Yiannis S. Chatzizisis.

**Investigation:** Wei Wu, Anastasios Nikolaos Panagopoulos, Aleem Siddique, Yiannis S. Chatzizisis.

**Methodology:** Wei Wu, Mohammadali Sharzehee, Aleem Siddique, Yiannis S. Chatzizisis.

**Project administration:** Wei Wu, Aleem Siddique, Yiannis S. Chatzizisis.

**Resources:** Wei Wu, Mohammadali Sharzehee, Aleem Siddique, Yiannis S. Chatzizisis.

**Software:** Wei Wu, Anastasios Nikolaos Panagopoulos, Mohammadali Sharzehee, Aleem Siddique, Yiannis S. Chatzizisis.

**Supervision:** Wei Wu, Aleem Siddique, Yiannis S. Chatzizisis.

**Validation:** Wei Wu, Anastasios Nikolaos Panagopoulos, Mohammadali Sharzehee, Shijia Zhao, Saurabhi Samant, Usama M. Oguz, Behram Khan, Abdallah Naser, Khaled M. Harmouch, Ghassan S. Kassab, Aleem Siddique, Yiannis S. Chatzizisis.

**Visualization:** Wei Wu, Mohammadali Sharzehee, Shijia Zhao, Saurabhi Samant, Usama M. Oguz, Behram Khan, Abdallah Naser, Khaled M. Harmouch, Ghassan S. Kassab, Aleem Siddique, Yiannis S. Chatzizisis.

**Writing – original draft:** Wei Wu, Anastasios Nikolaos Panagopoulos, Mohammadali Sharzehee, Shijia Zhao, Saurabhi Samant, Usama M. Oguz, Behram Khan, Abdallah Naser, Khaled M. Harmouch, Ghassan S. Kassab, Aleem Siddique, Yiannis S. Chatzizisis.

**Writing – review & editing:** Wei Wu, Anastasios Nikolaos Panagopoulos, Charu Hasini Vasa, Mohammadali Sharzehee, Shijia Zhao, Saurabhi Samant, Usama M. Oguz, Behram Khan, Abdallah Naser, Khaled M. Harmouch, Ghassan S. Kassab, Aleem Siddique, Yiannis S. Chatzizisis.

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
