## [Decision Letter · Decision Letter 0]

5 Aug 2022

PONE-D-22-17881Patient-Specific Computational Simulation of Coronary Artery Bypass GraftingPLOS ONE

Dear Dr. Chatzizisis,

Thank you for submitting your manuscript to PLOS ONE. After careful consideration, we feel that it has merit but does not fully meet PLOS ONE’s publication criteria as it currently stands. Therefore, we invite you to submit a revised version of the manuscript that addresses the points raised during the review process.

We look forward to receiving your revised manuscript.

Kind regards,

Adélia Sequeira, Ph.D

Academic Editor

PLOS ONE

Journal Requirements:

"Yiannis S. Chatzizisis: Speaker honoraria, advisory board fees and research grant from Boston Scientific Inc., Advisory board fees and research grant from Medtronic Inc., U.S. patent (No. 21072P) for the invention entitled “Patient-specific computational planning of coronary artery bypass grafting”, Co-founder of ComKardia Inc. All other authors have no relevant conflict of interests to disclose."

Additional Editor Comments:

The manuscript has several weaknesses and needs an extensive and careful revision.

I agree with the comments made by the reviewers and recommend the authors to take into account those comments and submit an improved version of the manuscript.

Reviewers' comments:

Reviewer's Responses to Questions

**Comments to the Author**

1. Is the manuscript technically sound, and do the data support the conclusions?

Reviewer #1: Partly

Reviewer #2: Partly

2. Has the statistical analysis been performed appropriately and rigorously? 

Reviewer #1: N/A

Reviewer #2: No

3. Have the authors made all data underlying the findings in their manuscript fully available?

Reviewer #1: Yes

Reviewer #2: Yes

4. Is the manuscript presented in an intelligible fashion and written in standard English?

Reviewer #1: Yes

Reviewer #2: Yes

5. Review Comments to the Author

Reviewer #1: The paper deals with a CFD study for coronary by-passes. In particular, the authors compute the Fractional FLow reserve (FFR) for two patients and compare it with angiography measures with good results.

The paper is of interest for the journal, however some issues should be better discussed by the authors in the text before publication:

MAJOR REMARKS:

1. Introduction. The literature about computational studies for coronary by-passes should be improved, I suggest for example to include:

- Nordgaard H, Swillens A, Nordhaug D, Kirkeby-Garstad I, Van Loo D, Vitale N, Segers P, Haaverstad R, Lovstakken L, Impact of competitive flow on wall shear stress in coronary surgery: computational fluid dynamics of a LIMA-LAD model. Cardiovasc Res, 2010

- Guerciotti B., Vergara C., Ippolito S., Quarteroni A., Antona C., Scrofani R., Computational study of the risk of restenosis in coronary bypasses. Biomechanics and Modeling in Mechanobiology, 2017

2. Related to the previous point: collocate the paper with respect to the previous works done on the topic: which is the novelty, what is common to other works? In general the first part of Introduction is too technical and long, I would rather focus on contestualization of the paper.

3. a) To complete the state of the art about the topic, authors should also mention the works that focused on FSI for coronary by-passes, e.g. I suggest to include:

- Kabinejadian F, Ghista DN (2012) Compliant model of a coupled sequential coronary arterial bypass graft: effects of vessel wall elasticity and non-newtonian rheology on blood flow regime and hemodynamic parameters distribution. Med Eng Phys, 2012

-Guerciotti B., Vergara C., Ippolito S., Quarteroni A., Antona C., Scrofani R., A computational fluid-structure interaction analysis of coronary Y-grafts. Medical Engineering & Physics, 2017

b) Moreover, the authors should discuss in the Limitations section the assumption of rigid walls

4) "Aim 1: To describe and test a new patient-specific computational framework"

In which sense "new"? Here and later in the Methods the authors should be more precise on this point

5) Methods: "... computationally created focal lumen stenoses with four degrees of severity (mild, moderate,severe, critical)"

Please cite previous works where different degrees of stenosis were virtually created, e.g

- Guerciotti B., Vergara C., Ippolito S., Quarteroni A., Antona C., Scrofani R., Computational study of the risk of restenosis in coronary bypasses. Biomechanics and Modeling in Mechanobiology, 2017

6) Page 8: "The boundary condition values were tuned until ..."

In which sense "bc values"? What is tuned? The values of parameters (R,C,...)? Please provide more details on the calibration phase and report the final parameters values used in the simulations.

7) The role of TAWSS and OSI should be better discussed in terms of clinical relevance. Usually low TAWSS and high OSI refer to regions where restenosis may occur after by-pass implantation

Reviewer #2: In this paper the authors employ a sophisticated computational model to investigated the hemodynamics before and after CABG the main claim being that the procedure "faithfully reproduces the hemodynamic effects of bypass grafting on the native coronary artery flow".

I must say that I find the study interesting, though not particularly original, even if there are several fundamental questions that should be answered and points to clarify before deciding about the suitability for publication of this paper.

A first relevant point is that the authors extensively use the adjective "patient specific" when referring to their simulations while they should honestly recognise that they are only loosely related to the real patients.

In fact, the geometries are obtained by "manually" segmenting the lumen boundaries using an available software; this step alone introduces uncertainties and any other operator wold produce a slightly different geometry.

Furthermore the authors state that "We assumed that only 4% of the cardiac output was distributed to the coronary arteries with a 70% and 30% split in the flow between the left and the right coronary artery, respectively." These are standard data known from the literature and it is very unlikely that the real patients from which the geometries have been extracted have exactly these proportions of the flows.

Another choice that seems quite irrational (although it is probably suggested by computational complexity) is to consider the lumen wall rigid and then couple this non deformable geometry with Windkessel-like boundary conditions

which include the compliance of the missing network of veins and arteries. If I have to think about a tract of the circulation where deformability is key is certainly the initial part of the aorta in which the blood pressure increased during systole is stored into elastic energy of the deformed walls.

Finally, also the hemodynamic parameters are not specific of the two patients.

I have also important reservations about the validation of the results: the FFR values of figure 4 have little meaning if the level of confidence in those numbers is not assessed. My experience is that pressure changes sharply downstream of a stenosis and numbers can vary a lot by slightly changing the sampling position(s).

The fact that the simulations are run using 2000 time steps per heartbeat, using some millions of elements, suggests that the method is numerically stabilised in some way (artificial viscosity, upwind discretizations) but there is no mention of numerical details.

To conclude, the study might be interesting but, as is, there are too many unclear points, unjustified statements and bold claims to recommend publication.

6. PLOS authors have the option to publish the peer review history of their article (what does this mean?). If published, this will include your full peer review and any attached files.

Reviewer #1: No

Reviewer #2: No

---

## [Author Response · Author response to Decision Letter 0]

6 Oct 2022

Reviewer 1: We would like to thank the Reviewer for the critique. We provided detailed clarifications and revisions based on the Reviewer suggestions and comments. 

Reviewer 2: We would like to thank the Reviewer for the critique. We provided detailed clarifications and revisions based on the Reviewer suggestions and comments.

---

## [Decision Letter · Decision Letter 1]

1 Nov 2022

PONE-D-22-17881R1

Patient-Specific Computational Simulation of Coronary Artery Bypass Grafting

PLOS ONE

Dear Dr. Chatzizisis,

Thank you for submitting your manuscript to PLOS ONE. After careful consideration, we have decided that your manuscript does not meet our criteria for publication and must therefore be rejected.

Specifically:

There is no clear reference to previous existing literature on the topic. The manuscript does not provide a sound scientific study adequate for publication.

I am sorry that we cannot be more positive on this occasion, but hope that you appreciate the reasons for this decision.

Kind regards,

Adélia Sequeira, Ph.D

Academic Editor

PLOS ONE

Additional Editor Comments:

The authors didn't take into account the comments made by the reviewers and did not answer to their questions. The new version of the manuscript was not sufficiently improved.

Reviewers' comments:

Reviewer's Responses to Questions

**Comments to the Author**

1. If the authors have adequately addressed your comments raised in a previous round of review and you feel that this manuscript is now acceptable for publication, you may indicate that here to bypass the “Comments to the Author” section, enter your conflict of interest statement in the “Confidential to Editor” section, and submit your "Accept" recommendation.

Reviewer #1: (No Response)

Reviewer #2: (No Response)

2. Is the manuscript technically sound, and do the data support the conclusions?

Reviewer #1: Partly

Reviewer #2: Partly

3. Has the statistical analysis been performed appropriately and rigorously? 

Reviewer #1: N/A

Reviewer #2: No

4. Have the authors made all data underlying the findings in their manuscript fully available?

Reviewer #1: Yes

Reviewer #2: No

5. Is the manuscript presented in an intelligible fashion and written in standard English?

Reviewer #1: Yes

Reviewer #2: No

6. Review Comments to the Author

Reviewer #1: I am not able to find the detailed answers of authors to my points

I am not able to find the detailed answers of authors to my points

Reviewer #2: I am sorry to have to say that the author have made no efforts to answer my questions and counter my criticisms.

They have add a few vague sentences here and there without really amending the flaws of the study.

Also the answer to my report given in the form "Reviewer 2: We would like to thank the Reviewer for the critique.

We provided detailed clarifications and revisions based on the Reviewer suggestions and comments." (identical

to that for Reviewer 1) confirms the reduced effort made for the revision.

My recommendation is to reject the paper.

7. PLOS authors have the option to publish the peer review history of their article (what does this mean?). If published, this will include your full peer review and any attached files.

Reviewer #1: No

Reviewer #2: No

- - - - -

---

## [Author Response · Author response to Decision Letter 1]

1 Dec 2022

Reviewer 1: We would like to thank the Reviewer for the critique. We provided detailed clarifications and revisions based on the Reviewer suggestions and comments.

Reviewer 2: We would like to thank the Reviewer for the critique. We provided detailed clarifications and revisions based on the Reviewer suggestions and comments.

---

## [Decision Letter · Decision Letter 2]

24 Jan 2023

Patient-Specific Computational Simulation of Coronary Artery Bypass Grafting

PONE-D-22-17881R2

Dear Dr. Chatzizisis,

We’re pleased to inform you that your manuscript has been judged scientifically suitable for publication and will be formally accepted for publication once it meets all outstanding technical requirements.

Kind regards,

Redoy Ranjan, MBBS, MRCSEd, Ch.M., MS (CV&TS), FACS

Academic Editor

PLOS ONE

Additional Editor Comments (optional): The topic of the study is essential, and the authors provide an interesting analysis. However, as CCTA is not superior to invasive coronary angiography, acknowledge its study limitation, and highlight the applicability of their model to clinical practice.

Review Comments to the Author

Reviewer #3: "Patient-Specific Computational Simulation of Coronary Artery Bypass Grafting" is a very interesting and innovative paper well written with high perspectives in therms of fluid dynamics and physical computational analysis, i suggest the publication in this current form.

Congratulation to the Authors!

Reviewer #4: This study, designed to test a new tool in the field of coronary artery disease. But there is a significant limitation to this study that should be mentioned.

CCTA is not superior to invasive coronary angiography in clearing the anatomy of the coronary arteries. No CABG could be performed based only on CCTA. This is because the resolution of invasive angiography is superior to that of CCTA and the images from CCTA are not good enough to take a patient to the OR. So, any products based on CCTA could not replace a coronary angiography. Also, it is a common sense, that an increase in stenoses would increase the graft flow and vice versa. Another point is, FFR was presented as a separate procedure, however, FFR is an optional part of coronary angiography. This means, if you do coronary angiography, you can add FFR measurement to the procedure but you can not do FFR without coronary angiography!

From viewpoint of a cardiac surgeon, I cannot see any new aspects in this paper that has been uncovered. But a software specialist may find it interesting.

Reviewer #5: This is a novel piece of work on computational framework that combines CCTA with CFD to calculate the local hemodynamics in native coronary arteries and bypass grafts. Moreover, the authors aimed to investigate the effect of bypass grafting on the local hemodynamics in the native artery and competitive flow between the graft and native coronary artery. To demonstrate that the proposed computational framework is applicable in the clinical setting, they performed a multifactorial computational analysis accounting for resting and hyperemic conditions and varying degrees of native coronary artery stenosis.

Overall it is a well written paper but there are certain minor points.

- The introduction should be significantly truncated and be focused solely on the objective of this paper

- The authors should highlight the applicability of their model to clinical practice

- Splitting the paper in two Aims -albeit appropriate for a grant proposal- is rather confusing for the reader so integrating all primary and secondary aims to a main objective would be preferred.

Reviewer #6: Dear the authors of the manuscript entitled "Patient-Specific Computational Simulation of Coronary Artery Bypass Grafting"

Thank you for taking in consideration all the reviewers comments

I have no concerns about this manuscript in its current status

---

## [Editor Report · Acceptance letter]

23 Feb 2023

PONE-D-22-17881R2 

Patient-Specific Computational Simulation of Coronary Artery Bypass Grafting 

Dear Dr. Chatzizisis:

I'm pleased to inform you that your manuscript has been deemed suitable for publication in PLOS ONE. Congratulations! Your manuscript is now with our production department. 

Kind regards, 

on behalf of

Dr. Redoy Ranjan 

Academic Editor

PLOS ONE